# Key Role of Phosphorylation in Small Heat Shock Protein Regulation via Oligomeric Disaggregation and Functional Activation

**DOI:** 10.3390/cells14020127

**Published:** 2025-01-17

**Authors:** Zachary B. Sluzala, Angelina Hamati, Patrice E. Fort

**Affiliations:** 1Department of Ophthalmology & Visual Sciences, The University of Michigan, Ann Arbor, MI 48109, USA; zsluzala@umich.edu (Z.B.S.); hamatia@umich.edu (A.H.); 2Department of Molecular & Integrative Physiology, The University of Michigan, Ann Arbor, MI 48109, USA

**Keywords:** sHSP, HSPB1, HSPB4, αA-crystallin, HSPB5, αB-crystallin, phosphorylation, post-translational modification, PTM

## Abstract

Heat shock proteins (HSPs) are essential molecular chaperones that protect cells by aiding in protein folding and preventing aggregation under stress conditions. Small heat shock proteins (sHSPs), which include members from HSPB1 to HSPB10, are particularly important for cellular stress responses. These proteins share a conserved α-crystallin domain (ACD) critical for their chaperone function, with flexible N- and C-terminal extensions that facilitate oligomer formation. Phosphorylation, a key post-translational modification (PTM), plays a dynamic role in regulating sHSP structure, oligomeric state, stability, and chaperone function. Unlike other PTMs such as deamidation, oxidation, and glycation—which are often linked to protein destabilization—phosphorylation generally induces structural transitions that enhance sHSP activity. Specifically, phosphorylation promotes the disaggregation of sHSP oligomers into smaller, more active complexes, thereby increasing their efficiency. This disaggregation mechanism is crucial for protecting cells from stress-induced damage, including apoptosis, inflammation, and other forms of cellular dysfunction. This review explores the role of phosphorylation in modulating the function of sHSPs, particularly HSPB1, HSPB4, and HSPB5, and discusses how these modifications influence their protective functions in cellular stress responses.

## 1. Introduction

Heat shock proteins (HSPs) are stress-induced proteins found in various organisms, including bacteria, plants, and animals [1]. They function as chaperones, preventing protein misfolding and aiding in protein refolding in stress conditions [2]. Small heat shock proteins (sHSPs), ranging from 12 to 43 kDa, are a subgroup of HSPs classified as HSPB1–HSPB10 [3,4]. Among them, HSPB1, HSPB4, and HSPB5 underlie much of our understanding of sHSPs. HSPB1 was the first bona fide sHSP to be identified [5,6], while HSPB4 and HSPB5 have been particularly studied due to their essential roles in maintaining the transparency and refractive properties of the eye lens [7,8]. HSPB6 has also been the subject of investigation due to its roles, along with HSPB1, in muscle contraction and relaxation (more specific reviews on sHSP phosphorylation in muscle contraction/relaxation can be found at [9,10,11,12]).

sHSPs are characterized by a conserved 80−100 amino acid sequence called the ‘⎱-crystallin core’ or ‘⎱-crystallin domain’ (ACD) [13,14,15,16,17,18,19], which is made up of beta strands forming an Ig fold [20,21,22]. This core is flanked by flexible but less well-conserved N- and C-terminal extensions [23] (see Figure 1 for an overview of these regions and the protein motifs). sHSPs generally form dimers, which then assemble into tetramers or hexamers, ultimately leading to the formation of large, heterogenous homo-oligomers [18,24,25,26,27,28], as well as hetero-oligomers with other sHSPs [29]. Dimerization is mainly mediated by interactions within the ACD [21,29,30,31,32,33,34,35,36,37,38,39,40,41,42,43,44], while further oligomerization from dimers into tetramers is primarily driven by the C-terminal extension, particularly the well-conserved I/V-X-I/V motif [1,31,45,46]. The N-terminal domain plays a key role in forming large oligomeric complexes, with the somewhat conserved WDPF [47] and (S/G)RLFD [48,49] motifs playing significant roles.

sHSP structure and oligomeric profile are tightly linked to their chaperone function, with smaller oligomeric species understood to generally have a greater capacity to chaperone substrate proteins due to increased surface area and exposure of otherwise buried hydrophobic patches. This illustrates the importance of the typically heterogeneous oligomeric profile and the necessity of a finely tuned sHSP chaperone system. At the structural level, the ACD, C-terminal extension, and N-terminal domain are each implicated in regulating sHSP chaperone and protective function. Within the C-terminal extension, the I/V-X-I/V motif seems to be involved in regulating the pH- and temperature-dependency of ⎱-crystallin chaperone function. Mutant forms of both HSPB4 and HSPB5 in which the hydrophobic isoleucine and valine residues of the I/V-X-I/V motif are mutated into glycine show improved chaperone function at 25 °C when compared to WT proteins [45]. Additionally, in lower pH environments, hydrophobic substrates have less competition from the I/V-X-I/V motif for binding sites [41]. Within the N-terminal domain, the well-conserved (S/G)RLFD motif and surrounding residues seem to be critical in maintaining normal chaperone abilities. Several studies have shown that removal of the (S/G)RLFD motif impacts sHSP function, generally resulting in defective chaperone and/or protective capabilities [50,51,52,53,54,55]. Mutation of this motif can also be detrimental, and several mutations on R21 of HSPB4, for example, which corresponds to the “R” of the HSPB4 (S/G)RLFD motif, have been associated with cataracts [56,57,58,59,60]. Interestingly, in the context of their role in preventing neurodegeneration, it is worth noting that several sHSP mutations have also been associated with various myopathies and neuropathies [61,62,63].

**Figure 1 cells-14-00127-f001:**
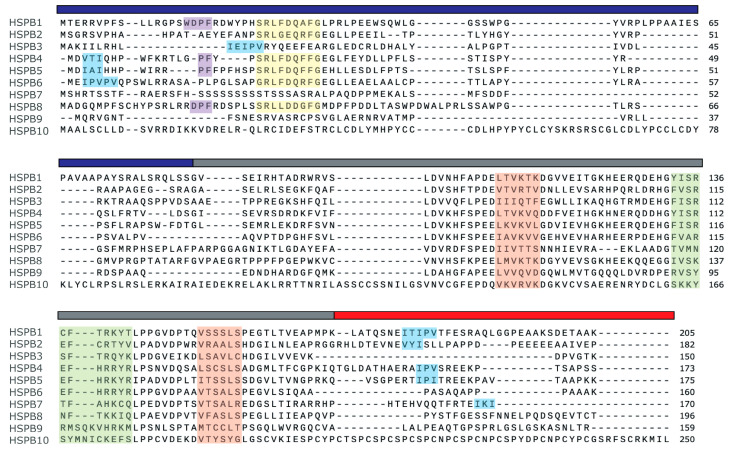
Human sHSP sequence alignment adapted from [64]. The colored bars above sequences represent the N-terminal domain (blue), the α-crystallin domain (ACD; gray), and the C-terminal domain (red). Highlighted amino acids represent relatively well-conserved motifs, including the I/V-X-I/V motifs (light blue), the WDPF motif (purple), the (S/G)RLFD motif (yellow), the β4 and β8 strands comprising the groove (orange), and the β6 + 7 strand comprising the dimer interface (green). Sequences were aligned with MUSCLE via SnapGene.

In addition to the intrinsic regulatory roles of various sHSP regions, extrinsic regulatory processes, such as post-translational modifications (PTMs), are also key modulators of their structure and function. Among the most studied sHSPs, HSPB4 and HSPB5 have garnered significant attention due to their involvement in cataract formation, providing a unique opportunity to explore PTMs in the aging lens. Cells of the lens do not turn over with age and instead migrate toward the center of the lens [65]. Similarly, lens proteins, especially sHSPs, are long-lived [66], allowing for the investigation of PTM associations with age. As cataracts also increase with age, they also provide a model for interrogating PTM associations with disease conditions. HSPB4 and HSPB5 have been shown to undergo increasing levels but also a broad range of modifications during aging, with different PTMs having a variety of impacts on their structure and function [67].

Deamidation is among the most reported modifications affecting HSPs and has often been shown to be detrimental to sHSP function. This type of modification generally increases molecular mass, resulting in larger oligomers, reduced solubility, and reduced surface hydrophobicity, which impairs chaperone activity [52,68,69,70]. Isomerization, which is less well-studied in part because it is more difficult to detect, has similarly been shown to have detrimental effects. sHSP isomerization has been associated with reduced solubility, altered oligomer size and stability, and age-related conditions such as cataracts [71,72]. Oxidation and disulfide bond formation are other PTMs reported as critical for regulating proper sHSP function [29,50,73]; however, excessive oxidation often leads to larger oligomers with reduced chaperone capacity, diminishing the protective function of sHSPs under stress conditions [74,75,76,77]. Acetylation and glycation have mixed and combinatorial effects. On its own, glycation can either increase, decrease, or have no impact on chaperone function, depending on the source of glycation, substrate, and context [77,78,79,80,81,82,83]. Acetylation alone can disrupt the native oligomeric assembly of α-crystallin [84], can increase or decrease chaperone function depending on substrate proteins [81,85], and can also modulate glycation-associated functional changes [83]. In the cataractous lenticular context, the deleterious impacts of several of these PTMs are evident. Excessive oxidation can lead to the generation of photosensitizers such as *N*-formykynurenine (NFK), kynurenine (KYN), hydroxytryptophans (HTRP), and H_2_O_2_ [86,87,88,89,90], and excessive glycation can result in the formation of advanced glycation end-products (AGEs), which in turn induce protein crosslinking and exacerbate the protein aggregation typical of a cataract [91].

Phosphorylation, in contrast, stands out as a particularly significant PTM, having a profound and often beneficial impact on sHSP structure and function. This modification typically induces a shift in the oligomeric profile of sHSPs, promoting a transition from large aggregates to smaller, more active complexes with enhanced chaperone activity [92,93,94,95,96,97,98,99,100,101,102]. Phosphorylation at specific serine or threonine residues modulates the protein’s ability to interact with substrate proteins, thereby increasing its capacity for stress-induced protection, such as in oxidative stress, ischemia, and thermal stress. Previously identified phosphosites on human HSPB1, HSPB4, HSPB5, and HSPB6 can be found in Table 1. The intricate regulation of sHSPs via phosphorylation underscores their critical role in fine-tuning the cellular response to environmental stressors, thereby contributing to cellular homeostasis and survival. Due to our current understanding of sHSPs, this review focuses on the role of HSPB1, HSPB4, and HSPB5 phosphorylation in regulating their structure and function, with additional information on HSPB6 also presented. While outside of the scope of this review, sHSP phosphorylation is also highly implicated in cytoskeletal regulation. For reviews more specific to those functions, see [103,104].

## 2. Regulation of HSPB1 by Phosphorylation

Human HSPB1 is phosphorylated primarily on S15, S78, and S82. Phosphorylation on all three of these sites is mediated via two separate regulatory pathways. “Pathway A” involves activation of p38 MAPK, which leads to phosphorylation of S15, S78, and S82 by MAPKAPK2 [106,127], MAPKAPK3 [128,129,130], and MAPKAPK5 [131]. “Pathway B” involves phosphorylation by PKC-δ [132] and to a lesser extent PKC-α [133,134].

Phosphorylation at these sites leads to a significant structural transition in HSPB1, facilitating the dissociation of larger aggregates into smaller complexes [92,93,94,95,96,97,98]. Studies using phosphomimetic and non-phosphorylatable mutant forms of HSPB1 validate this. The non-phosphorylatable triple alanine (3A) mutant of HSPB1 forms predominantly large oligomers, while the phosphomimetic triple aspartic acid (3D) mutant is characterized by an overrepresentation of small oligomers and dimers [92,93,94,95,135,136], as well as increased sensitivity to pH-induced structural changes [137] and alterations in secondary structure [136]. Single and double phosphomimetic HSPB1 mutants reveal that these alterations are amplified upon phosphorylation at multiple sites. Single mutants (S15D, S78D, S82D) exhibit only slightly altered secondary structure but little to no oligomeric disaggregation, while double mutants (S15D/S78D, S15D/S82D) exhibit more highly altered secondary structure and somewhat reduced oligomeric size [136]. This effect culminates in the triple phosphomimetic mutant (3D), which predominantly exists as dimers. This same additive disaggregation process is not seen in all other species. For example, hamster HSPB1 (phosphorylatable on S15 and S90) exhibits an oligomeric profile biased towards small complexes upon double phosphomimetic mutation (S15E/S90E) or S90E single mutation, but a profile biased towards large complexes upon either double non-phosphorylatable alanine mutation (S15A/S90A), S90A single mutation, or S15E single mutation [47,138]. S15A hamster HSPB1, while primarily forming large oligomers, also disaggregates upon S90 phosphorylation in vitro [47]. This suggests that disaggregation of hamster HSPB1 is more specifically dependent on S90 phosphorylation rather than dependent on and amplified by phosphorylation on multiple sites, as is the case for human HSPB1.

Consistent with smaller sHSP oligomers being more functionally active, phosphorylation generally enhances HSPB1 chaperone and protective capabilities. Accordingly, 3D (or triply phosphorylated WT) HSPB1 has been shown to more effectively chaperone α-lactalbumin [94], BSA [136], κ-casein [136], and insulin [94,136] than WT HSPB1. At least for insulin, this functional increase appears temperature-dependent, as 3D HSPB1 has been shown to have improved chaperone function at either 37 or 45 °C [94,136] but separately to have reduced chaperone function relative to WT HSPB1 at 30 °C [93]. As with the dissociation findings presented above, phosphorylation on multiple sites seems crucial for the full functional enhancement of HSPB1. Double phosphomimetic mutants (S15D/S78D, S15D/S82D, S78D/S82D) exhibit similarly enhanced chaperone function for insulin and BSA relative to WT HSPB1 [136], while single phosphomimetic mutants either exhibit no improvement (S15D), or improvement relative to WT but reduced function relative to double/triple mutants. S78D shows a slight improvement in insulin assays, while S82D shows a slight improvement in both insulin and BSA assays [136]. These trends align with phosphorylation-induced structural disaggregation enabling better substrate interactions.

The intricacies of phosphorylation-mediated impacts are further illustrated when investigating HSPB1 chaperone function in protective contexts; 3D, but neither 3A, S15A, nor S82A HSPB1 has been shown to protect against oxygen and glucose deprivation-induced cell death [139]. Of note, this study did not investigate the impacts of single phosphomimetic mutation or combinatorial double mutants. HSPB1 thermoprotective capabilities are also phosphorylation-dependent. The doubly phosphomimetic S15E/S90E and S15D/S90D mutant hamster HSPB1 forms provide increased thermoprotection, while the non-phosphorylatable S15A/S90A mutant shows reduced thermoprotective capacity [138]. Unlike the specifically S90-regulated disaggregation, dual phosphorylation appears necessary for full protection, as even singly phosphomimetic mutants S15E/S90A or S15A/S90E exhibit reduced thermoprotection [138]. Furthermore, phosphorylation plays a critical role in the regulation of HSPB1 anti-apoptotic properties. Phosphorylated HSPB1 exerts cytoprotective effects by targeting key apoptotic signaling pathways. For instance, phosphorylated HSPB1 suppresses mitochondrial Bax translocation, mitochondrial cytochrome c release, and PUMA upregulation, with the 3D phosphomimetic mutant being particularly effective in these roles, whereas the non-phosphorylatable 3A mutant fails to provide this protection [139]. However, 3A HSPB1 has previously been shown to be protective against TNF-α-induced stress, while 3D HSPB1 was not [95]. This reflects a nuanced interplay between phosphorylation and stress specificity, though phosphorylation remains the dominant mechanism for functional activation.

In the context of neuronal ischemia, HSPB1 also demonstrates phosphorylation-dependent neuroprotective effects. Overexpression of wild-type HSPB1 or phosphomimetic forms (S15D, S78D, and S82D) inhibits apoptosis signal-regulating kinase 1 (ASK1) signaling pathways, which are crucial for mediating cell death in ischemic stress [139]. In contrast, non-phosphorylatable mutants (S15A, S78A, S82A) fail to confer neuroprotection or inhibit ASK1 activation. Phosphorylation at S15 and S82 has been identified as essential for these protective effects, though of note regulated by PKD in this context (as opposed to MAPKAP kinases or PKC) [139].

Phosphorylation emerges as a central mechanism governing HSPB1 function. By inducing disaggregation into smaller, active complexes, phosphorylation enhances chaperone activity, thermoprotection, and anti-apoptotic properties. While exceptions exist, the predominant trend reflects a phosphorylation-mediated activation process tightly linked to stress-responsive regulation.

## 3. Regulation of HSPB4 by Phosphorylation

Compared to HSPB1, much less is known about the structural effects of phosphorylation on HSPB4. Historically, S122 was identified as the sole bona fide phosphorylation site on HSPB4. While the specific kinases responsible for phosphorylation at this site have not been determined, its phosphorylation has been shown to be cAMP-dependent in the calf, rat, and rabbit [140,141], possibly implicating PKA. Our lab later revealed that S/T148 (threonine in humans, serine in other mammals) could also be phosphorylated in vivo [120], though it is unclear by which kinase(s). At least in vitro, T148 phosphorylation appears to be mediated by several kinases, potentially in a cell-specific manner. We have recently identified mTORC2 as a T148-phosphorylating kinase, as well as shown evidence of potential regulation by PIK3C2A (in retinal neurons), PFKP and CDK1 (in Müller glia), PRKD2, NEK9, MAP2K1, MAP2K2, MAP3K7, MAP4K4, RPS6KA3, and AKT1 [142].

The structural consequences of phosphorylation at these sites remain poorly characterized relative to our understanding of HSPB1. For example, following H_2_O_2_ exposure, rat HSPB4 phosphorylated on unspecified residues formed two oligomeric populations: one resembling the unphosphorylated state (~650 kDa) and another larger (~1300 kDa) [143]. These findings conflict with the trend observed in HSPB1 and in sHSPs more broadly, where phosphorylation is generally associated with disaggregation. However, the specific sites involved and other potential modifications were not considered, so confounds cannot be ruled out. More recently, our lab has explored the structural implications of T148 phosphorylation using phosphomimetic (T148D) and non-phosphorylatable (T148A) mutants. We discovered that T148D forms slightly smaller oligomers, while T148A forms slightly larger ones compared to WT HSPB4 [99]. These shifts align more closely with trends seen in HSPB1, where phosphorylation correlates with smaller oligomer sizes. Additionally, T148D exhibits reduced susceptibility to stress-induced insolubility, whereas T148A shows increased insolubility [99]. These findings reinforce the hypothesis that phosphorylation promotes structural alterations favoring stability and function, similar to trends observed in HSPB1, though more data are needed to generalize this relationship.

Also similar to HSPB1, phosphorylation enhances HSPB4 chaperone function and protective capacity. For example, in one study, phosphorylated HSPB4 demonstrated enhanced chaperone function for βL-crystallin [77]. Along with the structural findings presented above, our lab also demonstrated that phosphomimetic T148D (but not T148A) HSPB4 exhibits improved chaperone function for ADH [99]. Suggestive of potentially additive or synergistic impacts of multi-site phosphorylation, singly- and doubly-phosphomimetic rat HSPB4 (S122E, S148E, S122E/S148E) have been shown to be protective of astrocytes against C2-ceramide or staurosporine-induced astrocyte cell death, whereas the corresponding non-phosphorylatable mutants (S122A, S148A, S122A/S148A) were not [144]. In this study, the doubly phosphomimetic and non-phosphorylatable mutants, respectively, exhibited the greatest increase and decrease in protective capacity [144]. Our lab has also shown that WT and T148D (but not T148A) human HSPB4 are protective against retinal neuron cell death, as demonstrated by reduced DNA fragmentation, caspase 3/7 activity, and ER stress [99,120]. We have also demonstrated T148 phosphorylation-mediated prevention of stress-associated mitochondrial Bax translocation [145]. More recently, another study has shown that HSPB4-mediated protection of photoreceptors from FasL-induced cell death is more pronounced in T148D and less pronounced in T148A expressing cells, also corresponding to increased or decreased interaction with Faim2 [146]. Additionally, T148 phosphorylation plays a role in inflammation regulation, as WT and T148D—but not T148A—HSPB4 reduce stress-induced IL−6, IL−1β, MCP−1, and IL−18 levels [147]. These findings position HSPB4 phosphorylation as consistently protective across different contexts, contrasting the slight variability observed with HSPB1.

## 4. Regulation of HSPB5 by Phosphorylation

Phosphorylation of HSPB5 has been more extensively studied than HSPB4, revealing both shared trends with HSPB1 and unique complexities. HSPB5 is phosphorylated on S19, S45, and S59. S45 phosphorylation is primarily mediated by ERK1/2 (p44/42 MAPK) [148], while S59 phosphorylation is primarily mediated by MAPKAPK2 [148,149], illustrating some level of similarity between its regulation and that of HSPB1. Also similar to HSPB1, phosphorylation at these sites tends to reduce oligomeric size. Triple phosphorylated WT HSPB5, as well as phosphomimetic (3D and 3E) mutants, exhibit smaller oligomeric complexes compared to unphosphorylated WT HSPB5 [100,101,102]. Evidencing additive or synergistic impacts as with HSPB1, this disaggregation depends on the phosphorylation of multiple sites, as S19D and S19D/S45D double mutants exhibit relatively unchanged oligomeric size but increased polydispersity and disrupted dimeric substructure [150]. The 3E HSPB5 has also been associated with increased susceptibility to trypsin degradation and reduced heat-induced insolubility [102], while 3D HSPB5 has similarly been associated with decreased urea stability [101], further indicating their tendency to disaggregate.

The functional consequences of HSPB5 phosphorylation align with trends seen in HSPB1 and HSPB4 but with notable complexities and substrate-specific effects. In general, phosphorylation enhances HSPB5 chaperone activity. For instance, WT and phosphorylated HSPB5 exhibit improved aggregation prevention of βL-crystallin compared to non-phosphorylatable mutants [77,151]. Similarly, triple phosphomimetic 3E HSPB5 demonstrates increased chaperone function for MDH and p53 [102], and 3D HSPB5 exhibits improved chaperone function for insulin, CS, and α-synuclein [101]. These findings suggest that phosphorylation facilitates structural or functional changes that enhance HSPB5 protective capabilities.

However, HSPB5 exhibits distinct substrate- and site-specific phosphorylation effects. Exemplary of these intricacies is the HSPB5-FBX4 interaction. HSPB5 has been shown to interact with FBX4 and to promote FBX4-dependent ubiquitination of insoluble proteins in a phosphorylation-specific manner. S19D/S45D and 3D HSPB5 show interaction with FBX4, but neither other phosphomimetic (S19D, S45D, S59D, S45D/S59D, S19D/S59D) nor non-phosphorylatable mutants (S19A, S45A, S59A, or combinatorial ‘A’ mutants) exhibit this interaction [152]. This indicates that, at least in this context, S19 and S45 phosphorylation are both necessary for chaperone function. However, symptomatic of the complexity of the regulation of sHSP chaperone function, S19D, S45D, S19D/S45D, and 3D HSPB5 all exhibit an impaired ability to chaperone mutant transmembrane proteins Fz4-FEVR and ATP7B-H1069Q, while S59D, S19D/S59D, and S45D/S59D HSPB5 maintain the ability to chaperone them [153]. This suggests that in this context, S59 is the primary regulator of HSPB5 chaperone activity, while S19 and S45 phosphorylation might even impede interaction with the substrates.

While infrequent, there are instances where phosphomimetic HSPB5 mutants exhibit deleterious impacts as well. For example, 3E and S59E mutants are associated with increased vinblastine-induced apoptosis, enhanced interaction with Bcl−2, and greater mitochondrial translocation of Bcl−2, while non-phosphorylatable mutants exhibit reduced apoptosis and weaker Bcl−2 interactions [123]. This suggests that there are contexts in which increased protein–protein interaction through enhancement of the phosphomimetic form’s chaperone function can actually result in negative consequences. Additionally, 3E HSPB5 is more susceptible to cycloheximide-induced degradation compared to 3A mutants [154] and fails to protect against TRAIL-induced apoptosis or inhibit caspase−3 activation [155,156]. Other studies have found that S19D/S45D HSPB5 rapidly co-aggregates with α-lactalbumin and promotes precipitation [150], and that 3D HSPB5 exhibits reduced chaperone ability for LDH [100]. Collectively, these findings suggest that HSPB5 phosphorylation, while generally protective, can have context-specific deleterious effects depending on the signaling pathways and substrates involved.

## 5. Regulation of HSPB6 by Phosphorylation

While the structural and functional impacts of phosphorylation on other small heat shock proteins are less comprehensively studied than those of HSPB1, HSPB4, and HSPB5, there are notably similar effects worth mentioning, particularly regarding HSPB6. The primary site of phosphorylation for HSPB6 is S16, mediated by PKA [126,157,158,159,160,161,162,163], PKG [126,164], and PKD [165] (though MAPKAPK2 has also been shown to induce HSPB6 phosphorylation in vitro [158]). S16 phosphorylation similarly leads to the dissociation of large macromolecular aggregates [157,166], with the non-phosphorylatable S16A HSPB6 having a tendency to form larger oligomers as well [167]. Notably, increased levels of S16 phosphorylation have been observed in failing human hearts and in mouse models post-ischemic/reperfusion insult, and in fact hearts of transgenic mice expressing the S16A mutant HSPB6 have been shown to exhibit impaired recovery and increased necrosis and apoptosis following I/R, relative to non-TG hearts [167].

As with the other sHSPs, there are contexts in which the phosphomimetic S16D HSPB6 has been found to be functionally enhanced. For example, S16D HSPB6 exhibits increased interaction with amyloid-β, inhibiting the formation of both globular and fibrillar aggregates and, in turn, enhancing protection against cellular toxicity [168]. The S16D mutant also protects against β-agonist-induced apoptosis and reduces caspase−3 activity [169,170]. Of note, there are contexts in which the S16D mutant does not exactly recapitulate the properties of phosphorylated WT HSPB6. HSPB6 has indeed been demonstrated to be an effective chaperone for 14−3−3, with this interaction being contingent upon S16 phosphorylation [43,171,172]. However, neither the unphosphorylated WT HSPB6 nor S16D HSPB6 can effectively interact with 14−3−3, while the S16-phosphorylated WT can, the two forming what appears to be a functional complex [171]. This has implications for its chaperone function in other contexts, as both S16-phosphorylated and S16D HSPB6 have been shown to exhibit reduced chaperone function for insulin in the absence of this functional complex; but the WT form, through interaction with 14−3−3, can regain the ability to effectively chaperone insulin [171]. This illustrates the important point that while phosphomimetic and non-phosphorylatable mutant forms of sHSPs can serve as important experimental tools, they do not always fully model the impacts of phosphorylated and unphosphorylated WT proteins.

## 6. Conclusions

Collectively, these findings highlight the complex and varied roles of phosphorylation in modulating the structural and functional properties of different sHSPs, supporting that these modifications are crucial for their chaperone functions and protein interactions within the cellular context. Through the induction of oligomeric disaggregation, phosphorylation has the capacity to functionally enhance sHSPs and increase their protective potential. This has profound implications in terms of the potential use of sHSPs in therapeutic capacities [173,174,175]. Our understanding of sHSP phosphorylation could be strengthened by a specific investigation of HSPB3/7/8/9/10 PTMs, about which virtually nothing is known at this time. Further, the findings presented above regarding oligomeric disaggregation largely pertain to homo-oligomerization; however, the impacts of phosphorylation on sHSP hetero-oligomerization [29] remain poorly understood. Finally, while we briefly discuss the regulation of various sHSP phosphosites by protein kinases, little is known regarding their regulation by phosphatases [176,177,178,179,180], providing another important avenue for future research.

It should be noted that sHSPs are subject to a wide variety of PTMs besides phosphorylation, each of which results in different impacts on sHSP structure and function. While this review and several of the sources referenced focus specifically on phosphorylation alone, the possible presence and impact of other modifications should not be discounted and may explain some of the divergent disease- or condition-specific findings presented above. For example, possible relationships between sHSP isomerization and phosphorylation have been discussed elsewhere [71], and combinatorial modification could lead to a variety of different impacts. There are obviously other sHSP PTMs that have been shown to generally result in activating effects as well, one of which worth mentioning in this review is O-GlcNAcylation since it similarly targets serine and threonine residues. sHSP O-GlcNAcylation has been associated with increased chaperone function, larger and more heterogenous oligomers, and altered interactomes (including altered interactions with other sHSPs) [181,182,183]. Further study of post-translational regulation of sHSP properties is thus warranted to interrogate the possibility of harnessing their chaperone and protective roles for disease prevention or management, particularly of “activating” PTMs such as phosphorylation or O-GlcNAcylation.

## Figures and Tables

**Table 1 cells-14-00127-t001:** Previously identified phosphosites on the human HSPB1, HSPB4, HSPB5, and HSPB6.

Human sHSP	Previously Identified Phosphosite(s)
HSPB1	S15 [105,106,107,108]; S78 [105,106,107,108,109,110]; S82 [105,106,107,108,109,110,111,112,113,114]
HSPB4	T13 [112,115,116]; S20 [115]; T43 [115]; S45 [115,116,117]; Y47 [115]; S51 [115]; T55 [115]; S59 [115]; S62 [115]; S66 [112,115]; S81 [112,115]; T86 [115]; Y118 [115]; S122 [115,116,117,118,119]; S127 [115]; S130 [115]; T140 [116]; T148 [112,115,120]; T153 [112,115]; S162 [115]; T168/S169 [115]; S172 [115]; S173 [115]
HSPB5	S19 [105,108,112,115,116,117,119,121,122]; S21 [112,115,116,121]; S43 [115,116]; S45 [105,108,115,116,117,119,122,123]; S53 [115,116]; S59 [105,108,112,115,116,117,119,121,122,123,124]; T63 [115]; S66 [115]; S76 [112,115,116]; S85 [115]; T132 [115]; T134 [115]; S136 [115]; S138 [115]; S139 [112,115]; S153 [115]; T158 [115]
HSPB6	S16 [125,126]

## Data Availability

The original contributions presented in this study are included in the article. Further inquiries can be directed to the corresponding author(s).

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
