# Peer review of "Key Role of Phosphorylation in Small Heat Shock Protein Regulation via Oligomeric Disaggregation and Functional Activation"

_cells, 2025, doi:10.3390/cells14020127_

Round 1
Reviewer 1 Report
Comments and Suggestions for Authors
The authors have prepared a review of phosphorylation’s impact on small heat shock protein (sHSP) structure and function. While other reviews have explored various aspects of sHSP biology, this review’s focus on phosphorylation is welcome because recent reviews on this particular sub-topic are not as broad (e.g. Bakthisaran et al. 2016 BBA-General Subjects 1860:167-182; Muranova et al. 2018 Biochemistry (Moscow) 83:1196-1206 and Thornell and Aquilina 2015 Cell Mol Life Sci. 72:4127-4137). The article is well written and clearly referenced. The organizational structure is also clear and the focus on HSPB1, HSPB4 and HSPB5 is justified. This article is recommended for publication, with a few suggested changes:
(1) Page 2, line 72-73. In addition to cataracts, it might be worth briefly mentioning that HSPB4 and HSPB5 mutations are associated with cardiomyopathies and motor neuropathies (Ranek et al. 2017 Philos. Trans. Soc. Lond. B. Biol. Sci. 373:20160530).
(2) Page 3, line 95: define NFK, KYN, HTRP.
(3) Matt Pratt’s group has reported ways of synthesizing uniformly O-GlcNac modified HSPB1 (Balana et al. 2021 Nat Chem. 13:441-450; Moon et al. 2023 ACS Chem. Biol. 18:1705-1712; Javed et al. 2024 Prot. Sci. 33:e5173). Those studies have shown that O-GlcNac also regulates HSPB1 structure and function. He has also mentioned potential interplays between phosphorylation and O-GlcNac, which might be important in this current article. Finally, Pratt’s review on targeting sHSP PTMs could also be a useful parallel to the current work (Wang and Pratt 2024 Trends. Pharm. Sci. 45:583-585), with a more translational focus.
(4) Pseudo-phosphorylation of HSPB1 has also been shown to promote binding to the microtubule-associated protein tau and suppression of its aggregation (Freilich et al. 2018 Nat Comm. 9:4563; Baughman et al. 2018 J. Biol. Chem. 293:2687-2700). This result might be an interesting addition because tau is intrinsically disordered, unlike other substrates that have been mentioned.
(5) Other PTM-related citations to maybe consider adding: Deamidation effects on HSB4 solubility (Willmarth et al. 2008 J. Proteome Res. 5:1554-2556. Presence of L-isoAsp in HSPB4 (Lyon et al. 20918 Expo. Eye Research 131-141; Lambeth et al. 2019 ACS Central Sci. 5:1387-1395.
(6) In the Discussion, it might be worth specifically noting some of the major gaps in the field, with the goal of inspiring young scientists to work on these problems. For example, there is virtually nothing known about PTMs on HSPB3, 7, 8, 9 and 10. The authors mention this in the body of the review, but specifically calling for action in the Discussion might better highlight this gap. There is also very little known about the phosphatases that act on HSPBs. Finally, it might be worth mentioning again that sHSPs form heteroligomers in vivo (the authors already cite: Mymrikov et al. 2012 Cell Stress Chaperones 17:157), and little is known about how phosphorylation specifically impacts this process.
Author Response
Q. Page 2, line 72-73. In addition to cataracts, it might be worth briefly mentioning that HSPB4 and HSPB5 mutations are associated with cardiomyopathies and motor neuropathies (Ranek et al. 2017 Philos. Trans. Soc. Lond. B. Biol. Sci. 373:20160530).
A: The text has been modified to include a brief note about sHSP mutations in myopathies and neuropathies (bolded text is newly added) “Mutation of this motif can also be detrimental, and several mutations on R21 of HSPB4 for example, which corresponds to the “R” of the HSPB4 (S/G)RLFD motif, have been associated with cataract [56-60]. Interestingly, in the context of their role in preventing neurodegeneration, it is worth noting that several sHSP mutations have also been associated with various myopathies and neuropathies [61-63].” (LINES 66-71)
Different references than the one suggested were included due to their more specific focus on sHSPs rather than on HSPs, and their more extensive cataloging of specific involved mutations.
Q. Page 3, line 95: define NFK, KYN, HTRP.
A: These definitions have been added.
Q. Matt Pratt’s group has reported ways of synthesizing uniformly O-GlcNac modified HSPB1 (Balana et al. 2021 Nat Chem. 13:441-450; Moon et al. 2023 ACS Chem. Biol. 18:1705-1712; Javed et al. 2024 Prot. Sci. 33:e5173). Those studies have shown that O-GlcNac also regulates HSPB1 structure and function. He has also mentioned potential interplays between phosphorylation and O-GlcNac, which might be important in this current article. Finally, Pratt’s review on targeting sHSP PTMs could also be a useful parallel to the current work (Wang and Pratt 2024 Trends. Pharm. Sci. 45:583-585), with a more translational focus.
A: The suggested citations have been added, with a brief overview of the impacts of O-GlcNAcylation in the discussion (bolded text is newly added): “There are obviously other sHSP PTMs which have been shown to generally result in activating effects as well, one of which worth mentioning in this review being O-GlcNAcylation since it similarly targets serine and threonine residues. sHSP O-GlcNAcylation has been associated with increased chaperone function, larger and more heterogenous oligomers, and altered interactomes (including altered interactions with other sHSPs) [181-183].” (LINES 372-377). The suggested review article has also been added to the citation list, along with other reviews on the therapeutic potential of sHSPs.
Q. Pseudo-phosphorylation of HSPB1 has also been shown to promote binding to the microtubule-associated protein tau and suppression of its aggregation (Freilich et al. 2018 Nat Comm. 9:4563; Baughman et al. 2018 J. Biol. Chem. 293:2687-2700). This result might be an interesting addition because tau is intrinsically disordered, unlike other substrates that have been mentioned.
A: sHSP interactions with cytoskeletal proteins are certainly critical examples of their chaperone function, however, due to the vast amount of work done regarding these interactions, we have opted to keep the present review’s focus on other substrates and functions, instead citing several reviews which readers can turn to for a more specific and detailed overview. See lines 124-126, “While outside of the scope of this review, sHSP phosphorylation is also highly implicated in cytoskeletal regulation. For reviews more specific to those functions, see [103,104].”
Q. Other PTM-related citations to maybe consider adding: Deamidation effects on HSB4 solubility (Willmarth et al. 2008 J. Proteome Res. 5:1554-2556. Presence of L-isoAsp in HSPB4 (Lyon et al. 20918 Expo. Eye Research 131-141; Lambeth et al. 2019 ACS Central Sci. 5:1387-1395.
A: Several of the suggested citations have been added. Specifically, the following text was added (bolded text is newly added): “Deamidation is among the most reported modifications affecting HSPs and has often been shown to be detrimental to sHSP function. This type of modification generally increases molecular mass, resulting in larger oligomers, reduced solubility, and reduced surface hydrophobicity, which impairs chaperone activity [52,68-70]. Isomerization, which is less well-studied in part because it is more difficult to detect, has similarly been shown to have detrimental effects. sHSP isomerization has been associated with reduced solubility, altered oligomer size and stability, and age-related conditions such as cataract [71,72].” (LINES 89-96).
Q. In the Discussion, it might be worth specifically noting some of the major gaps in the field, with the goal of inspiring young scientists to work on these problems. For example, there is virtually nothing known about PTMs on HSPB3, 7, 8, 9 and 10. The authors mention this in the body of the review, but specifically calling for action in the Discussion might better highlight this gap. There is also very little known about the phosphatases that act on HSPBs. Finally, it might be worth mentioning again that sHSPs form heteroligomers in vivo (the authors already cite: Mymrikov et al. 2012 Cell Stress Chaperones 17:157), and little is known about how phosphorylation specifically impacts this process.
A: The reviewer makes some very important points and we have now incorporated them in this review with the following changes. The following brief note has been added to the introduction to mention hetero-oligomerization (bold text is newly added): “sHSPs generally form dimers which then assemble into tetramers or hexamers, ultimately leading to formation of large, heterogenous homo-oligomers [18,24-28], as well as hetero-oligomers with other sHSPs [29].” The following text has also been added to the discussion to address these areas of future research (bold text is newly added): “Our understanding of sHSP phosphorylation could be strengthened by specific investigation of HSPB3/7/8/9/10 PTMs, about which virtually nothing is known at this time. Further, the findings presented above regarding oligomeric disaggregation largely pertain to homo-oligomerization, however, the impacts of phosphorylation on sHSP hetero-oligomerization [29] remain poorly understood. Finally, while we briefly discuss the regulation of various sHSP phosphosites by protein kinases, little is known regarding their regulation by phosphatases [176-180], providing another important avenue for future research.” (LINES 355-363).
Reviewer 2 Report
Comments and Suggestions for Authors
Under stressor circumstances that adversely impact protein structure, a network of players orchestrates a molecular stress response to maintain protein homeostasis. The main actors in the stress response are molecular chaperones, which comprise the heat shock protein family. Among them, small heat shock proteins, sHPSs, can be found in every realm of life. The key features of sHSPs can be summarized as follows: a) they are ATP-independent chaperones; b) they are defined by their small molecular weights (12-43 kDa) and c) they exist in ensembles of oligomeric species differing in size.
Besides the intrinsic regulatory roles of the different sHSPs domains, key modulators of sHSPs structure and function are extrinsic regulatory processes, represented by post-translational modifications (PTMs), including deamidation, acetylation, glycation and phosphorylation. The latter has been shown to trigger a profound and beneficial impact on the sHSPs' structure and function by promoting the transition from large aggregates to smaller and more active complexes displaying enhanced chaperone activity.
In the review titled "Key role of phosphorylation in small heat shock protein regulation via oligomeric disaggregation and functional activation" Sluzala and colleagues discuss the role of phosphorylation in modulating the function of some sHSPs, namely HSPB1, HSPB4, HSPB5 and HSPB6.
Overall, the manuscript offers a comprehensive view of the topic.
Minor amendments required before publication include:
a) Figure 1 is not mentioned in the main text. Either mention it or remove the "Figure 1";
b) a couple of typos seem scattered throughout the main text (e.g. in Table 1 T43 should be in bold);
c) an additional figure depicting a model for the chaperone function of the sHPs, in which is summarized the equilibrium interconverting monomers versus oligomers, would help the readers who are not too familiar with the topic and thus strengthen the manuscript;
d) please double check the formatting of the references because it seems not following the journal guidelines.
Author Response
Q. Figure 1 is not mentioned in the main text. Either mention it or remove the "Figure 1";
A: The text has been modified to include a reference to Figure 1 (bolded text is newly added): “This core is flanked by flexible, but less well-conserved N- and C-terminal extensions [23] (see Figure 1 for an overview of these regions and the protein motifs).” (LINES: 41-42).
Q. a couple of typos seem scattered throughout the main text (e.g. in Table 1 T43 should be in bold);
A: We sincerely apologize for that. T43 has been bolded and the manuscript has been thoroughly double checked for typos.
Q. an additional figure depicting a model for the chaperone function of the sHPs, in which is summarized the equilibrium interconverting monomers versus oligomers, would help the readers who are not too familiar with the topic and thus strengthen the manuscript;
A: We really appreciate the reviewer’s suggestion and have added a new figure encompassing this as part of the graphical abstract for the review.
Q. please double check the formatting of the references because it seems not following the journal guidelines.
A: It has been double checked that the MDPI Endnote style has been used for references.